# First Single-Centre Experience with the Novel HIF-α Inhibitor Belzutifan in Switzerland

**DOI:** 10.3390/curroncol32020064

**Published:** 2025-01-26

**Authors:** Tobias Peres, Stefanie Aeppli, Stefanie Fischer, Thomas Hundsberger, Christian Rothermundt

**Affiliations:** 1Department of Medical Oncology and Hematology, Cantonal Hospital St. Gallen (KSSG), 9000 St. Gallen, Switzerland; stefanie.aeppli@kssg.ch (S.A.); stefanie.fischer@kssg.ch (S.F.); thomas.hundsberger@kssg.ch (T.H.); 2Department of Medical Oncology and Cancer Centre, Cantonal Hospital Lucerne (LUKS), 6000 Lucerne, Switzerland; christian.rothermundt@luks.ch

**Keywords:** belzutifan, HIF-α inhibitor, von Hippel–Lindau syndrome, renal cell carcinoma, haemangioblastomas, case series, real-world

## Abstract

Belzutifan is a new HIF-α inhibitor mainly used in two different indications: von Hippel–Lindau syndrome-associated renal cell carcinoma, haemangioblastomas and pancreatic neuroendocrine tumours, as well as sporadic advanced pre-treated renal cell carcinoma. Although efficacy has been demonstrated in phase II and III studies, belzutifan is still not approved in many countries. In addition, von Hippel–Lindau syndrome is a rare disease. Therefore, there is virtually no real-world experience data of belzutifan efficacy available. We aim to determine the real-world efficacy and tolerability of belzutifan in patients with von Hippel–Lindau syndrome-associated tumours and in patients with sporadic advanced tyrosine kinase- and immune checkpoint inhibitors pre-treated for renal cell carcinoma. A retrospective analysis of five patients treated with belzutifan between 2023 and 2024 at a Swiss cancer centre was conducted. In this case series, all patients consistently benefitted from belzutifan with response to treatment. This case series provides real-world evidence that belzutifan is an effective and well-tolerated treatment option for patients with von Hippel–Lindau syndrome-associated renal cell carcinoma, haemangioblastomas and sporadic advanced pre-treated renal cell carcinoma.

## 1. Introduction

### 1.1. Belzutifan: Mechanism of Action and Approval

Belzutifan is a novel hypoxia inducible factor 2 alpha (HIF-2α) inhibitor. As a transcription factor, HIF-2α has a role in gene regulation depending on the presence of hypoxia. In the presence of normal oxygen saturation, HIF-2α is directed by the VHL protein to ubiquitin-dependent proteasomal degradation. In the event of VHL protein deficiency, this leads to stabilisation and the accumulation of HIF-2α. Consecutively, HIF-2α migrates into the nucleus and interacts with HIF 1 beta (HIF-1b), forming a complex that subsequently leads to the expression of genes associated with cell proliferation, angiogenesis and tumour growth. Belzutifan binds to HIF-2α and blocks the HIF-2α-HIF-1b interaction, resulting in reduced transcription and expression of HIF-2α target genes [1]. The dosage for belzutifan in all indications is 120 mg daily, orally [2]. Belzutifan has recently been approved in Switzerland for the treatment of adult patients with Von Hippel–Lindau Syndrome (VHL)-associated renal cell carcinoma (RCC), central nervous (CNS) haemangioblastomas (HB) or pancreatic neuroendocrine tumours (pNET) for whom immediate surgery is not required. This approval is based on data from the phase II LITESPARK Study 004, which included 61 patients with VHL-associated RCC [3]. Some of the patients had other VHL-associated tumours, such as haemangioblastomas and pNETs, and could therefore be investigated. The response rate for RCC was around 60%, for pNET, 90% and for CNS haemangioblastomas, approximately 40%. Responses appear to be durable, as the respective median duration of response was not reached in the last update of the trial [4]. Depending on the type of response measurement (solid/cystic portion of the lesion), belzutifan even achieved significantly higher response rates in CNS HB in a further analysis [5]. In addition, the United States Food and Drug Administration (FDA) has already approved belzutifan in December 2023 for patients who have advanced RCC and have already received therapy with programmed death receptor-1 (PD-1) or programmed death-ligand 1 (PD-L1) inhibitors and a vascular endothelial growth factor tyrosine kinase inhibitor (VEGF-TKI). This approval is based on data from the phase III LITESPARK 005 trial [6], which randomised 746 patients with advanced clear cell (cc) RCC to either belzutifan or everolimus. Patients had to have previous exposure to a PD-(L)1 inhibitor and VEGF-TKI. The primary endpoint, progression-free survival (PFS), was significantly prolonged with belzutifan (HR 0.75; *p* = 0.0008). Numerically, there appears to be an overall survival (OS) advantage, but this was not statistically significant. The secondary endpoint objective response rate (ORR) was approximately 22% for belzutifan compared to 3% for everolimus. Table 1 provides an overview of the LITESPARK 004 and LITESPARK 005 studies. The rationale for the efficacy of belzutifan in ccRCC (independent of a VHL) is that most ccRCC tumours possess genetic or epigenetic alterations of the VHL gene [7,8].

### 1.2. Introduction VHL: Characteristic Tumours

Von Hippel–Lindau disease (VHL) is an autosomal dominant tumour predisposition syndrome caused by a genetic aberration of the VHL gene on chromosome 3. The disease usually manifests in young adults; the estimated prevalence of VHL in Europe is about 1–10/100,000. The increased occurrence of various tumours is associated with VHL. Among these, the most common tumours are retinal angiomas, CNS HBs, renal and pancreatic cysts, ccRCC, phaeochromocytoma and pNET. The lifetime risk of RCC and haemangioblastomas in VHL is very high, at 60–80% in each case [9].

Due to the restricted marketing approval, clinical experience with belzutifan in Switzerland is quite limited. However, following a large VHL cohort at our hospital, five of these patients with or without VHL are currently being treated with belzutifan. This article describes our experiences so far.

## 2. Detailed Case Description

Table 2 provides an overview of the five cases, with patient characteristics described below.

### 2.1. Belzutifan in Pre-Treated Advanced Renal Cell Carcinoma

Two patients at our Cancer Centre are currently receiving belzutifan for advanced ccRCC. Both patients received immunotherapy and a VEGF TKI before belzutifan.

**Case 1** A 77-year-old male patient with an initial diagnosis of localised ccRCC in 2018 and a corresponding nephrectomy had a systemic recurrence with pulmonary metastases in July 2020. Given the overall low volume of the disease and favourable risk (International Metastatic RCC Database Consortium (IMDC) 0 points), the decision was made at the time to conduct active surveillance [10]. Due to bilateral pulmonary progression in January 2021, systemic therapy with pembrolizumab and axitinib was started [11], and led to a good response. Due to severe fatigue, headaches, wound healing disorders and hepatitis G2, the TKI axitinib we stopped after only 3 months of therapy, hence pembrolizumab alone was continued. In February 2024, the patient developed new brain metastases and several left cervical lymph node metastases, which also led to high-grade internal jugular vein compression. Stereotactic radiotherapy was applied to the four brain metastases and the patient was switched to second-line therapy with the TKI cabozantinib. Due to concerns regarding tolerability, the patient received a reduced dose of cabozantinib (40 mg). However, the treatment had to be discontinued after about 4 weeks after the patient experienced headaches, dizziness and non-healing wounds again. After cost approval was obtained for belzutifan, treatment was initiated at the end of May 2024. A PET/CT scan and MRI head scan before the start of belzutifan treatment showed pulmonary progression as well as haemorrhaged and necrotic cerebral metastases with increasing oedema, most likely as a result of radiotherapy. After just 6 weeks of treatment with belzutifan, the CT scan from showed a good pulmonary response with clear regression of the metastases (see Figure 1 and Figure 2). The cervical and hilomediastinal lymph node metastases were slightly reduced in size. Only one of the four cerebral metastases was still definable. The current restaging from the end of October shows a sustained response. The patient tolerates belzutifan relatively well. He complains of mild headaches and slight dizziness. The haemoglobin level has dropped from >100 g/L before the start of therapy to 85 g/L in August. As per the recommendations, the dosage of belzutifan was reduced to 80 mg daily [2].

**Case 2** A currently 81-year-old female patient underwent nephrectomy in 2006 due to a localised ccRCC on the right side. An isolated right retrocaval lymph node metastasis was resected in 2014 and a new metastasis in the tail of the pancreas in 2021. Histopathologically, the pancreatic metastasis corresponded to a ccRCC. We decided to observe bilateral pulmonary metastases due to the relatively low tumour burden. In January 2023, multiple hepatic metastases were detectable. Systemic therapy with pembrolizumab and axitinib [11] was initiated for IMDC intermediate risk, which showed a good treatment response with complete remission of the hepatic metastases after only 4 months. However, pembrolizumab and axitinib were stopped due to immune-related hepatitis, which recovered on steroids. When single agent axitinib was restarted in July 2023 due to pulmonary and hepatic disease progression, acute liver failure occurred in August 2023, most likely as a drug-induced liver injury. Due to intolerance, both to immunotherapy and TKI, a cost approval for belzutifan was obtained from the patient’s health insurer. Belzutifan was started in April 2024, and the restaging in June 2024 showed pulmonary and hepatic tumour regression. However, due to worsening anaemia (down to 90 g/L), and also with a decrease in oxygen saturation, the dose of belzutifan was reduced to 80 mg daily in the meantime. Substrate deficiency was ruled out. The patient denied having any other possible side effects of belzutifan. The most recent restaging scans from November 2024 show stable hepatic and partly further regressive pulmonary metastases, resulting in an ongoing response overall.

### 2.2. Belzutifan in Von Hippel–Lindau-Associated Renal Cell Carcinoma

**Case 3** A 36-year-old male patient with non-familial VHL was treated at our centre for years. As the typical phenotype, he suffered from multiple organ involvement of diverse tumours. In addition to multifocal ccRCC and multiple spinal, cerebral and retinal HBs, he showed cystadenomas of both epididymides, an endolymphatic sac tumour (ELT) of the left temporal bone, and multiple renal and pancreatic cysts. He has been blind in his left eye since 2011 due to retinal angiomatosis. In 2012, a partial labyrinthectomy with excision of the ELT was performed. In 2013 and 2016, he received partial renal resections on the right side due to RCC. In 2020, a suboccipital craniotomy and resection of a HB of the brain stem were also necessary.

MR staging at the end of 2023 had shown progression of the multiple RCCs on the left, with the largest finding measuring almost 30 mm in extent, so that there was a considerable risk of development of metastases. In addition, the aim was to stabilise the renal tumours to avert the risk of dialysis following a renewed loss of kidney tissue due to surgical resection.

After receiving reimbursement, treatment with belzutifan was started at the end of November 2023. After 10 weeks of treatment with belzutifan, the MR staging showed a response of the RCC with size regression in almost all carcinomas. The largest tumour was now only 20 mm in size (Figure 3 and Figure 4). After a further nine months of treatment with belzutifan, the RCCs continued to be at least stable, although there was a slight overall trend towards a further reduction in size. The largest finding now measures 18 mm. Notably, the cerebral and spinal HBs were partly regressive or at least stable, while the cystic findings in the pancreas and kidneys were unchanged. The patient’s tolerance of belzutifan has always been excellent. There is no relevant anaemia or hypoxia.

### 2.3. Belzutifan in Von Hippel–Lindau-Associated Haemangioblastomas

**Case 4** A 20-year-old female patient with familial VHL has been treated with belzutifan for the indication of a large and growing spinal HB for 12 months. The initial finding was a right retinal angioma, which was managed repeatedly with laser coagulation. There are also two pancreatic cysts and a solitary renal cyst on the left. A right cerebellar HB was diagnosed in 2014 and resected in 2017. Multiple cystic haemangioblastomas in the cervical region have been known since 2014. In 2022, a multilocular progression became apparent in the spine, which was a large cystic-solid finding at the cervicothoracic junction extending from the fifth cervical to the fourth thoracal vertebrae and increasing medullary oedema. There were no neurological motor or sensory deficits for the patient. However, the cervicothoracic tumour complex represented an impending spinal cord compression. The neurosurgeons had already refrained from a resection due to the high risk of postoperative neurological disability. After 10 weeks of treatment with belzutifan, MR staging showed a response of the large cystic-solid finding in the cervicothoracic region (previously 12.5 cm in craniocaudal diameter, now 11.5 cm). Fortunately, perifocal oedema also regressed. Cerebellar HBs were stable. Haemoglobin level only fell slightly from 145 g/L to 125 g/L, and the patient did not develop any other side effects. The therapeutic benefit of belzutifan in this patient was confirmed after 6 months. Apart from mild belzutifan-induced anaemia, the patient tolerated the treatment without any side effects. After 11 months of treatment, the cystic-solid spinal HB progressed, with increasing perifocal oedema. However, it later transpired that the patient had not taken belzutifan for several weeks without giving any specific reasons or any new side effects. After only four weeks of taking belzutifan again, a short-term reduction in HBs was again demonstrated. Adherence to belzutifan is now reliable.

### 2.4. Belzutifan in Haemangioblastomatosis

**Case 5** A 60-year-old male patient presented with a symptomatic, solitary cerebellar HB. Surgical resection was performed with complete remission of neurological symptoms. A multimodal clinical evaluation did not reveal any evidence for VHL-disease. Clinically, he presents with disseminated haemangioblastomatosis. A somatic VHL mutation was repeatedly detected in resected specimens by molecular genetics. However, no germline mutation could be found. The first surgical intervention was performed in 2015 with the removal of a cerebellar HB. In the following years, various other resections of HBs were performed in the cranial and spinal regions. There have been no further manifestations to date. The present disseminated haemangioblastomatosis is considered an ultra-rare disease. Accordingly, there are no standard treatments, or evidence-based therapeutic recommendations that go beyond individual case reports. MR staging from May 2023 showed disease progression, both cerebrally and spinally. Most intracranial HBs were progressive in size, the largest of which was medially adjacent to the right amygdala region with progressive perifocal oedema. The vertebrospinal region also showed a progression in size in most known HBs (most prominent finding at the level of C3/4). Clinically, the patient reported mild headaches and neck pain, as well as mild ataxia, which had already existed in previous years and was not newly aggravated. An application for cost approval for belzutifan was rejected by the health insurance company in 2023. Following case reports, the VEGF inhibitor bevacizumab was administered in three weekly doses from mid-2023. After four cycles of bevacizumab, disease progression was initially halted. Unfortunately, a multilocular asymptomatic progression in the number and size of the intracranial and spinal HBs occurred as early as December 2023. Subsequently, the treatment interval with bevacizumab was intensified and shortened to 2 weeks. The patient also reported significantly worse hearing on the left side from the end of 2023, this hearing loss could also be objectified in an audiometry and was most likely due to compression of the auditory nerve on the left side by a growing HB. However, an MR restaging from March 2024 revealed further disease progression. Treatment with belzutifan was finally started in April 2024 after obtaining reimbursement. After about 10 weeks, belzutifan showed a substantial response in all cerebral and spinal findings. In some cases, the HBs almost halved in size (Figure 5 and Figure 6). The patient also benefited greatly from the treatment, subjectively. His hearing in his left ear improved noticeably within 4 weeks of starting treatment with belzutifan. He can now use his left ear to make a phone call again. In the meantime, his balance has also improved, and the patient is able to ride a bicycle again. From June, the patient developed mild anaemia under belzutifan (haemoglobin 120 g/L), asymptomatic for the patient. The current MRI again shows low size regression of isolated spinal HBs. The intracranial HBs are stable in size. Furthermore, no new spinal or intracranial HBs have occurred. Due to the ongoing response to treatment, both MRI graphically and subjectively, treatment is currently being continued unchanged with 120 mg belzutifan daily.

## 3. Discussion

In Switzerland, the HIF-2α inhibitor belzutifan has now been approved for VHL-associated RCC and CNS HBs. In the United States, belzutifan is also already approved for immunotherapy- and TKI-pre-treated advanced RCC.

In our five patients, we see the effect of belzutifan in the two mentioned indications, as well as in a patient with haemangioblastomatosis. All five patients consistently benefit from belzutifan, and the benefit has been sustained to date. It should also be emphasised that the treatment is very well-tolerated overall. No treatment interruptions were necessary in our patients. Anaemia is a typical side effect of belzutifan [12,13], and the dose therefore had to be reduced in two patients.

In patients with advanced RCC who have received multiple prior systemic therapies, a well-tolerated additional treatment option is of particular importance.

## 4. Limitations

Our experience with patients on belzutifan is subject to certain limitations. The cohort described here is small, heterogeneous and the follow-up is relatively short. The duration of treatment with belzutifan in our patients is currently between 9 and 15 months. There are a lack of long-term follow-up data from the studies that led to approval of belzutifan [3,6]. In our patients at least no short-term safety signals exceeding known toxicities from registration trials were observed. It is also unclear whether there are differences in the long-term efficacy of belzutifan in HB and RCC.

It should also be mentioned that there are also data and ongoing trials on the use of belzutifan in earlier lines of therapy. For example, the combination of belzutifan and cabozantinib showed promising activity in a phase II study in first-line advanced RCC [14]. The phase III LITESPARK-012 compares pembrolizumab plus lenvatinib, with or without belzutifan or quavonlimab (an anti-CTLA4-Ab) as first-line treatment in advanced ccRCC [15].

Belzutifan will probably move further forward in the therapy sequence in the future, as our patient experiences described above are, at least, encouraging.

## Figures and Tables

**Figure 1 curroncol-32-00064-f001:**
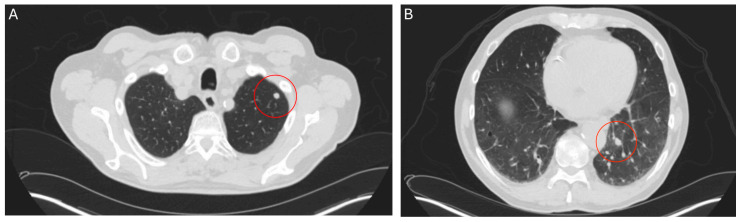
CT thorax scan, 27 May 2024, before commencement of treatment with belzutifan: pulmonary metastases for the left-upper (**A**) and lower lobe (**B**).

**Figure 2 curroncol-32-00064-f002:**
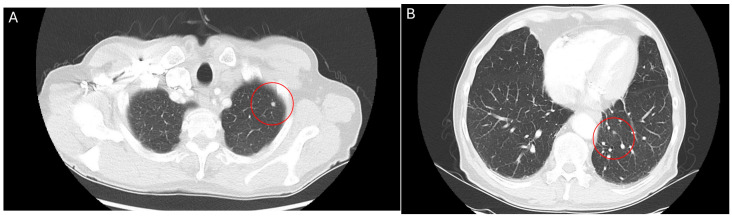
CT thorax scan, 12 July 2024, after 6 weeks treatment with belzutifan: regression of pulmonary metastases for the left-upper (**A**) and lower lobe (**B**).

**Figure 3 curroncol-32-00064-f003:**
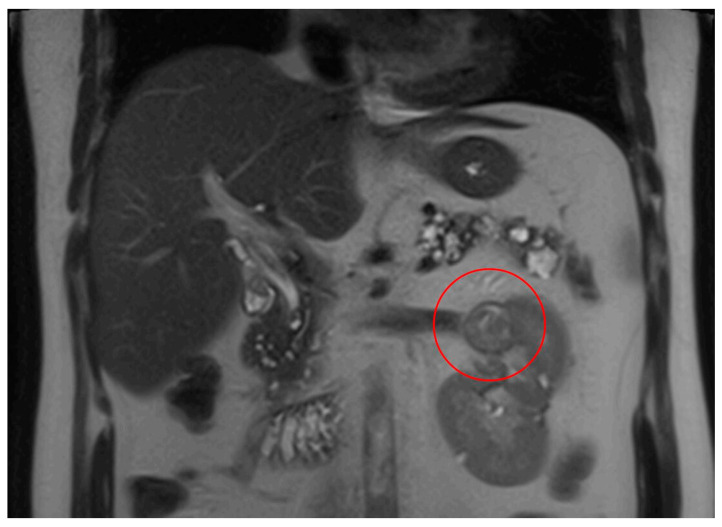
MRI abdomen scan, 21 November 2023, before commencement of treatment with belzutifan: multiple RCC left kidney (largest finding marked).

**Figure 4 curroncol-32-00064-f004:**
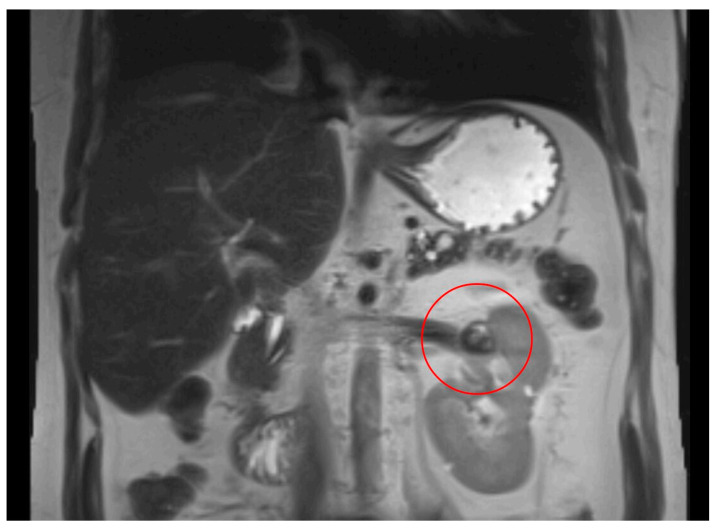
MRI abdomen scan, 6 February 2024, after 10 weeks treatment with belzutifan: regression of RCC.

**Figure 5 curroncol-32-00064-f005:**
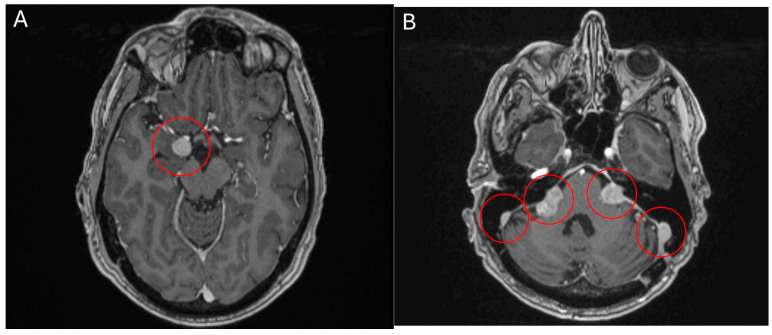
MRI head scan, 25 March 2024, before commencement of treatment with belzutifan: multiple CNS parasellar haemangioblastomas (**A**) and cerebellopontine angle (**B**).

**Figure 6 curroncol-32-00064-f006:**
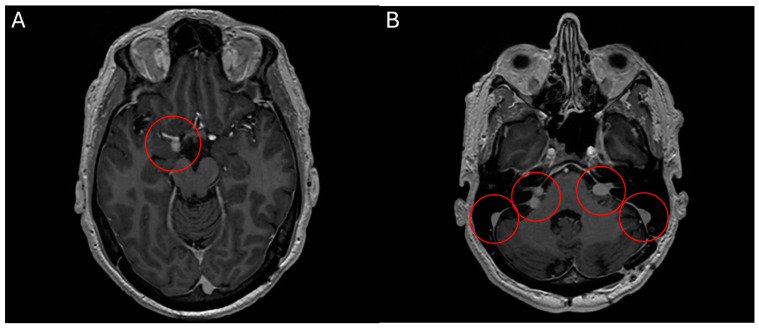
MRI head scan, 5 July 2024, after 10 weeks treatment with belzutifan: regression of CNS parasellar haemangioblastomas (**A**) and cerebellopontine angle (**B**).

**Table 1 curroncol-32-00064-t001:** Phase II and III trial data on belzutifan in VHL-associated RCC/pNET/HB and sporadic pre-treated ccRCC.

	Study Population	Study Design	Median Follow-Up (Months)	Efficacy Outcomes	Adverse Events (AE)
**LITESPARK004** [3,4]	VHL-associated RCC, pNET, CNS HB	Phase II, single-arm, belzutifan 120 mg once daily, until disease progression or unacceptable toxicity	29.3	ORR 59% (RCC, primary endpoint), 90% (pNET, secondary endpoint) 38% (CNS HB, secondary endpoint), median DOR (secondary endpoint) not reached	AE G3 16% (G3 anaemia 10%), no AE G4/5
**LITESPARK005** [6]	Sporadic advanced PD-(L)1 inhibitor and VEGF-TKI pre-treated ccRCC	Phase III, randomised, belzutifan 120 mg vs. Everolimus 10 mg once daily, until disease progression or unacceptable toxicity	18.4	OS (co-primary endpoint) 21.4 (belzutifan) vs. 18.1 (everolimus) months; PFS (co-primary endpoint) 5.6 (belzutifan) vs. 5.6 (everolimus) months; ORR (secondary endpoint) 21.9% (belzutifan) vs. 3.5% (everolimus)	AE G ≥ 3 61.8% (belzutifan) vs. 62.5% (everolimus)

ORR—overall response rate, OS—overall survival, PFS—progression-free survival, DOR—duration of response.

**Table 2 curroncol-32-00064-t002:** Patient characteristics.

	Age at Start of Belzutifan (Years)	Sites of Disease	Treatment Duration on Belzutifan (Months)	Tumour Response *	Toxicities **
**Case 1: Pre-treated ccRCC**	76	Pulmonary + brain metastases, cervical/thoracal lymph nodes metastases	9	Partial response	Anaemia G2, Headache G1, Dizziness G1
**Case 2: Pre-treated ccRCC**	81	Pulmonary + hepatic metastases	10	Partial response	Anaemia G2
**Case 3: VHL-associated RCC**	35	Multifocal ccRCC, intracranial and spinal HBs	15	Partial response	--
**Case 4: VHL-associated HBs**	19	Intracranial and spinal HBs	15	Stable disease	Anaemia G1
**Case 5: Haemangioblastomatosis**	60	Intracranial and spinal HBs	10	Partial response	Anaemia G1

* Tumour response (RECIST 1.1) according to investigator assessment. ** Grading (G) according to CTCAE 5.0, 2017.

## Data Availability

Due to privacy and ethical restrictions, the original personal data of the patients cannot be published.

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
