# Peer review of "First Single-Centre Experience with the Novel HIF-α Inhibitor Belzutifan in Switzerland"

_curroncol, 2025, doi:10.3390/curroncol32020064_

Round 1

Reviewer 1 Report

Comments and Suggestions for Authors

First single-centre experience with the novel HIF-α inhibitor Belzutifan in Switzerland 

Belzutifan, a HIF-α inhibitor, has shown efficacy in treating von Hippel Lindau syndrome-associated tumors and sporadic advanced pre-treated renal cell carcinoma, though it remains unapproved in many countries. The authors present a retrospective analysis of five patients treated at a Swiss cancer center between 2023 and 2024, demonstrating consistent benefits and good tolerability of the drug, albeit in a very small cohort. The authors aim to present a case series offering real-world evidence supporting belzutifan as an effective treatment option.

I have limited comments / suggestions. In the main, the manuscript reads well. There is quite extensive literature on the various LITESPARK trials based on a PubMed search. In my view, there could be a more in-depth review of the literature in the Introduction to provide some additional context to the reader, perhaps a table of the trials with an indication as to the main purpose of each trial (this is briefly mentioned to some extent in Discussion, but only for 012). It might also be useful to tabulate the baseline data for the five participants, and to have some commentary as to how they compare with (a) the range of individuals suffering these conditions and (b) the range of individuals enrolled in the various trials. It is difficult to assess from the text how generalisable these case studies might be, especially as the authors note, "It is remarkable that all five patients consistently benefit from belzutifan and that the benefit has been sustained to date" [line 249] which may indicate that the case series is not as representative of the larger trial performances. I think the final statement "Our patient experiences described above support at least its widespread use" might be an overstatement, I agree that the experiences do not provide any reason for concern, but I would not be comfortable extrapolating the outcomes of 5 patients to 'widespread use'.

Author Response

Comment 1: Belzutifan, a HIF-α inhibitor, has shown efficacy in treating von Hippel Lindau syndrome-associated tumors and sporadic advanced pre-treated renal cell carcinoma, though it remains unapproved in many countries. The authors present a retrospective analysis of five patients treated at a Swiss cancer center between 2023 and 2024, demonstrating consistent benefits and good tolerability of the drug, albeit in a very small cohort. The authors aim to present a case series offering real-world evidence supporting belzutifan as an effective treatment option.

I have limited comments / suggestions. In the main, the manuscript reads well. There is quite extensive literature on the various LITESPARK trials based on a PubMed search. In my view, there could be a more in-depth review of the literature in the Introduction to provide some additional context to the reader, perhaps a table of the trials with an indication as to the main purpose of each trial (this is briefly mentioned to some extent in Discussion, but only for 012). It might also be useful to tabulate the baseline data for the five participants, and to have some commentary as to how they compare with (a) the range of individuals suffering these conditions and (b) the range of individuals enrolled in the various trials. It is difficult to assess from the text how generalisable these case studies might be, especially as the authors note, "It is remarkable that all five patients consistently benefit from belzutifan and that the benefit has been sustained to date" [line 249] which may indicate that the case series is not as representative of the larger trial performances. I think the final statement "Our patient experiences described above support at least its widespread use" might be an overstatement, I agree that the experiences do not provide any reason for concern, but I would not be comfortable extrapolating the outcomes of 5 patients to 'widespread use'.

Response 1: 

  • We have responded to the request for the tables. We have included a table showing the 2 main relevant Phase II and III studies and a table with patient characteristics of the 5 patients.
  • Lines 258 and 281 have been modified with regard to the reviewers request

Reviewer 2 Report

Comments and Suggestions for Authors

The text is to long, especially description of five patients and have to be shortened, to be more concise and in the concordance  with figures.  

Author Response

Comment 2: The text is to long, especially description of five patients and have to be shortened, to be more concise and in the concordance  with figures.  

Response 2: 

  • We have responded to these suggestions and deleted a few sentences from the manuscript, where we thought it made sense to do so.

Reviewer 3 Report

Comments and Suggestions for Authors

The manuscript altogether seems to be an useful and timely contribution and provides new insight into the therapeutic potential of HIF-alfa inhibitor Belzutifan.

The manuscript contains a valuable collection of data of a first-center experience which were generated from the very limited series of human subjests. The study is potentially important, however the limited number of cases can make a limitation of the study.

After minor revision I would support accepting the manuscript.

Some specific comments:

Because the number of cases is small and somehow heterogeneous, please add more expanded section describing the limitatiion of the study.

I would add in the title: a pilot study

In conclusion the manuscript is undoubtedly interesting and fairly well written and the conceptual advance is pretty good thus it can be a welcome addition to the literature.

Author Response

Comments 3: 

The manuscript altogether seems to be an useful and timely contribution and provides new insight into the therapeutic potential of HIF-alfa inhibitor Belzutifan.

The manuscript contains a valuable collection of data of a first-center experience which were generated from the very limited series of human subjests. The study is potentially important, however the limited number of cases can make a limitation of the study.

After minor revision I would support accepting the manuscript.

Some specific comments:

Because the number of cases is small and somehow heterogeneous, please add more expanded section describing the limitatiion of the study.

I would add in the title: a pilot study

In conclusion the manuscript is undoubtedly interesting and fairly well written and the conceptual advance is pretty good thus it can be a welcome addition to the literature.

Response 3: 

  • In order to emphasize certain limitations of the work, we have created an additional paragraph (“4. Limitations”).
  • Thank you for the suggestion to call this a pilot-study. We feel this title would imply a more structured and prospective collection of patient data. This is in fact a case series. We would prefer to keep the title “First single-centre experience with the novel HIF-α inhibitor Belzutifan in Switzerland”.